# Efficient Photocatalytic Hydrogen Production over NiS-Modified Cadmium and Manganese Sulfide Solid Solutions

**DOI:** 10.3390/ma15228026

**Published:** 2022-11-14

**Authors:** Ksenia O. Potapenko, Evgeny Yu. Gerasimov, Svetlana V. Cherepanova, Andrey A. Saraev, Ekaterina A. Kozlova

**Affiliations:** Federal Research Center, Boreskov Institute of Catalysis SB RAS, Lavrentieva Ave. 5, 630090 Novosibirsk, Russia

**Keywords:** solid solution, NiS, Cd_1−x_Mn_x_S, photocatalysis, hydrogen production, visible light

## Abstract

In this work, new photocatalysts based on Cd_1−x_Mn_x_S sulfide solid solutions were synthesized by varying the fraction of MnS (x = 0.4, 0.6, and 0.8) and the hydrothermal treatment temperature (T = 100, 120, 140, and 160 °C). The active samples were modified with Pt and NiS co-catalysts. Characterization was performed using various methods, including XRD, XPS, HR TEM, and UV-vis spectroscopy. The photocatalytic activity was tested in hydrogen evolution from aqueous solutions of Na_2_S/Na_2_SO_3_ and glucose under visible light (425 nm). When studying the process of hydrogen evolution using an equimolar mixture of Na_2_S/Na_2_SO_3_ as a sacrificial agent, the photocatalysts Cd_0.5_Mn_0.5_S/Mn(OH)_2_ (T = 120 °C) and Cd_0.4_Mn_0.6_S (T = 160 °C) demonstrated the highest activity among the non-modified solid solutions. The deposition of NiS co-catalyst led to a significant increase in activity. The best activity in the case of the modified samples was shown by 0.5 wt.% NiS/Cd_0.5_Mn_0.5_S (T = 120 °C) at the extraordinary level of 34.2 mmol g^−1^ h^−1^ (AQE 14.4%) for the Na_2_S/Na_2_SO_3_ solution and 4.6 mmol g^−1^ h^−1^ (AQE 2.9%) for the glucose solution. The nickel-containing samples possessed a high stability in solutions of both sodium sulfide/sulfite and glucose. Thus, nickel sulfide is considered an alternative to depositing precious metals, which is attractive from an economic point of view. It worth noting that the process of photocatalytic hydrogen evolution from sugar solutions by adding samples based on Cd_1−x_Mn_x_S has not been studied before.

## 1. Introduction

Currently, most of the world’s energy is produced by burning raw fossil materials, which is an exhaustible natural resource [1]. Every year, there is a tendency of increasing temperatures, which is likely due to an increase in the amount of carbon dioxide in the atmosphere formed during the combustion of fossil fuels [2]. In this regard, transition to alternative energy sources is the most important task of modern society [3,4,5].

Renewable energy technologies include the use of biomass, solar and photovoltaic energy, hydropower, ocean thermal energy, and tidal energy [6]. In this case, solar energy is a strategically important resource, being the largest source of renewable energy available [7]. The most promising area for the development of solar energy is the direct conversion of light energy into the energy of chemical bonds [8]. An effective way in this case is to reproduce the functions of all types of natural photosynthesis by creating photocatalytic systems. The ultimate goal here is the photocatalytic production of hydrogen [9,10]. To achieve high efficiency of the photocatalytic hydrogen evolution process, it is necessary to develop materials that meet several requirements. The most important one is the efficiency of converting solar energy into the energy of chemical bonds under the action of visible light, which makes up approximately 43% of the solar spectrum [11,12,13,14]. Moreover, the cost and efficiency of the use of irradiation energy are important parameters from an economic point of view [15].

The Cd_1−x_Mn_x_S solid solution has recently attracted considerable attention because of its adjustable band gap depending on the manganese content and more negative position of the valence band compared with CdS [16,17,18]. Subsequent modification of the Cd_1−x_Mn_x_S surface makes it possible to significantly increase the rate of the process. The most common way to enhance the activity of solid solutions of cadmium and manganese sulfides is the deposition of noble metals, such as Ag [19] and Pt [20,21] or their compounds, on the surface of Cd_1−x_Mn_x_S. However, these metals are expensive, which is unsuitable for large-scale hydrogen production.

Among the previously studied co-catalysts, the precipitation of transition metal sulfides (such as NiS, CoS, WS_2_, and MoS_2_) [22,23,24,25,26,27,28,29,30] is of interest for their further application. The deposition of transition metal sulfides leads to a significant increase in the rate of hydrogen evolution due to the formation of a heterojunction, which contributes to the efficient separation of photogenerated charge carriers [22,23,24,25,26,27,28,29,30]. Previous authors [22,23,24] successfully synthesized a photocatalyst NiS/Cd_0.5_Mn_0.5_S. Earlier, it was been shown that the deposition of CuS on the Cd_0.3_Mn_0.7_S surface leads to a significant increase in the rate of hydrogen evolution [27,31].

For efficient hydrogen evolution, it is necessary to use electron donors—as sacrificial agents [32]. Electron donors are oxidized by photogenerated holes, thereby increasing the efficiency of the target process. An equimolar mixture of Na_2_SO_3_ and Na_2_S is one of the most common systems for hydrogen evolution [31]. From a practical point of view, it is considered interesting to use more accessible electron donors, such as polysaccharides from plant biomass, the main structural monomer of which is glucose [33]. When using biomass as electron donors, it is possible to obtain hydrogen using only renewable energy sources: water, solar irradiation, and biomass [33].

Previously, we proposed a method for the synthesis of active photocatalysts based on solid solutions of manganese and cadmium sulfides; the proposed technique included hydrothermal treatment at a temperature of 120 °C [34]. The aim of this work was to study the expansion of the range of hydrothermal treatment to 100–160 °C. In addition, the surface of the synthesized materials based on solid solutions of cadmium and manganese sulfides was modified with nickel sulfide. The deposition efficiency of the non-noble metal co-catalyst nickel sulfide was compared with a standard platinum co-catalyst. The proposed NiS/Cd_1−x_Mn_x_S photocatalysts were successfully tested in the production of hydrogen under visible light from aqueous solutions of inorganic and organic sacrificial agents—Na_2_S/Na_2_SO_3_ and glucose, respectively.

## 2. Materials and Methods

### 2.1. Photocatalyst Synthesis

The following reagents were used in the synthesis of photocatalysts: Mn(NO_3_)_2_·4H_2_O (Sigma-Aldrich, St. Louis, MO, USA, 97.0%), CdCl_2_·2.5H_2_O (Vekton, Krasnodar Krai, Russia, 98%), Na_2_SO_3_ (Acros organics, Geel, Belgium, 98%), NaOH (Sigma-Aldrich, St. Louis, MO, USA, 98%), Na_2_S (Biochem Chemopharma, Cosne-Cours-sur-Loire, France, 60%), Cu(NO_3_)_2_·2.5H_2_O (Acros organics, Geel, Belgium, 98%), Ni(NO_3_)_2_·6H_2_O (Sigma-Aldrich, St. Louis, MO, USA, 97%), HAuCl_4_ (Aurat, Moscow, Russia, 99%), and NaBH_4_ (Acros organics, Geel, Belgium, 98%).

#### 2.1.1. Cd_1−x_Mn_x_S Series

Photocatalysts Cd_1−x_Mn_x_S were prepared according to the method described in detail earlier [34]. To an aqueous solution of manganese nitrate 0.1 M Mn(NO_3_)_2_ was added a solution of 0.1 M CdCl_2_ in the required proportion and stirred for 30 min. Next, twofold excess of Na_2_S from the stoichiometric amount was added, and the resulting suspension was stirred for 30 min. The precipitation was washed and placed in an obturator-type autoclave. The hydrothermal treatment temperature was varied from 100 °C to 160 °C in steps of 20 °C with a heating temperature of 2 °C/min; autoclaves were kept at a given temperature for 24 h. The resulting precipitation was washed 4 times with distilled water, centrifuged, and dried at 70 °C for 4 h. The synthesized sample was designated as Mnx HTy, where x is the molar fraction of manganese in the solid solution, and y is the temperature of hydrothermal treatment.

#### 2.1.2. NiS/Mn0.6 HT120 Series

Nickel sulfide was deposited on the surface of the photocatalyst by ion exchange between nickel nitrate and sodium sulfide. Thus, Mn0.6 HT120 was suspended in 5–7 mL of distilled water, and then the proper amount of 0.1 M Ni(NO_3_)_2_ was added dropwise and continuously stirred for 1 h. Next, an excess of Na_2_S was added to the resulting suspension and stirred for 1 h. The precipitate was washed with distilled water 4 times, centrifuged, and dried at 70 °C for 1 h. Samples of 0.1, 0.3, 0.5, and 1 wt.% NiS/Mn0.6 HT120 were synthesized according to a similar scheme.

#### 2.1.3. Pt/Mn0.6 HT120 Photocatalyst

Sample was prepared by chemical reduction of platinum with sodium borohydride. Thus, 300 mg of Mn0.8 HT120 was suspended in 5–7 mL of distilled water, and then 304 μL of 0.05M H_2_PtCl_6_ was added dropwise and stirred for 1 h. An excess of NaBH_4_ solution was added to the resulting suspension and stirred for an hour. The precipitate was washed with distilled water 4 times and dried at 70 °C for 1 h.

### 2.2. Physical and Chemical Methods

The phase composition of the catalysts was studied by X-ray diffraction (XRD) on a Bruker D8 diffractometer (Bruker, Ettlingen, Germany) using CuKα irradiation. During the analysis, the angle 2θ was varied from 20 to 70° with a scanning step of 0.05°. The exact composition of the synthesized catalysts and the crystallite size were calculated using the TOPAS software. The x value in Cd_1−x_Mn_x_S was estimated on the basis of the linear dependence of the lattice stability and the concentration of manganese in the sample according to Vegard’s law. Hexagonal manganese sulfide and cadmium sulfide were chosen as reference points. The identification of the phase composition was carried out by comparing the experimentally obtained diffraction pattern with the diffraction pattern of the standard. The following cards were selected from the database of diffraction standards: MnS (PDF no. 401289), CdS (PDF no. 411049), β-Mn_3_O_4_ (PDF no. 270734), Mn(OH)_2_ (PDF no. 180787), and Pt (PDF no. 040802).

The UV-vis diffuse reflectance spectra were obtained with the use of Shimadzu UV-2501 PC (Shimadzu, Kyoto, Japan) spectrophotometer with an ISR-240A diffuse reflectance attachment in the wavelength range from 400 to 850 nm.

To determine the band gap of photocatalysts, the reflection spectra were recalculated into Kubelka–Munk coordinates:(1)F(R)=(1−R100)22R100,
where *R*(%) is the reflection coefficient. The optical band gap was estimated in Tauk coordinates by plotting (*F*(*R*)hv)^2^ against hv and linear extrapolation to the x-axis.

X-ray photoelectron spectroscopy (XPS) was used to study the chemical composition of the surface of the photocatalysts with the use of photoelectron spectrometer SPECS Surface Nano Analysis GmbH (Berlin, Germany) using non-monochromatized AlKα radiation (hυ = 1486.61 eV). Data processing was carried out using the CasaXPS software package. Transmission electron microscopy (TEM) was used to study the structure and microstructure of the catalysts using a ThemisZ electron microscope (Thermo Fisher Scientific, Waltham, MO, USA).

### 2.3. Photocatalytic Experiments

The activity of photocatalysts was determined in the reaction of photocatalytic hydrogen evolution from aqueous solutions of electron donors: a system of sodium sulfide and sulfite (0.1 M Na_2_S/0.1 M Na_2_SO_3_) and a glucose solution (220 mg α-D glucose, 400 mg NaOH). The experiments were carried out in a reactor with a quartz window with an area of 3.5 cm^2^ (Figure 1). An LED with a wavelength of 425 nm was used as an irradiation source.

A typical experiment can be described as follows: a 100 mL suspension containing 50 mg of catalyst was sonicated for 5 min and then placed in a reactor. Next, the reactor was purged with argon for 20 min to remove oxygen present in the system and then irradiated with visible light (425 nm, power 450 mW cm^−2^). The amount of evolved hydrogen was determined using a gas chromatograph KHROMOS GH-1000 (Russia) with a NaX zeolite column, and argon was a carrier gas.

The apparent quantum efficiency was calculated using the following formula:(2)$$AQE=\frac{2\cdot W\left(H_{2}\right)}{N_{\mathrm{photons}}}\cdot 100\%,$$
where *W*(*H*_2_) is the rate of photocatalytic hydrogen evolution (mol min^−1^), and N_photons_ (330 mmol photon min^−1^) is the number of photons of the incident radiation (photon min^−1^).

## 3. Results and Discussions

### 3.1. Photocatalyst Characterization

#### 3.1.1. XRD Analysis

The photocatalysts were studied by XRD. For all the samples, a general trend could be observed in that the XRD patterns contained three peaks characteristic of nanocrystalline cadmium and manganese sulfides (Figure 2).

For many of the samples, the XRD patterns showed a complex form due to the formation of various manganese compounds, such as β-Mn_3_O_4_, Mn(OH)_2_, and MnS. The formation of these phases was caused by the interaction of manganese cations with atmospheric oxygen and with sodium sulfide Na_2_S during the synthesis of photocatalysts. The structure of the solid solutions was predominantly hexagonal with a large number of stacking faults since some peaks were absent, and the ratio of the peak heights differed from the ideal hexagonal structure. As the manganese content increased, the particle size and unit cell volume decreased. Note that the unit cell volume for CdS is equal to 99.79 Å^3^ (PDF #41-1049), whereas for MnS, this parameter is equal to 88.40 Å^3^ (PDF #40-1289). In addition, there was an insignificant shift of the diffraction peaks toward larger angles 2θ, associated with the smaller radius of Mn^2+^ compared with Cd^2+^. Consequently, as x increased in Cd_1−x_Mn_x_S, the interplanar spacing decreased in the samples [27]. It should be noted (Table 1) that the parameter x in the solid solutions was close to that set during synthesis, despite the formation of impure manganese compounds. With an increase in the hydrothermal treatment temperature, as expected, the fraction of manganese cations introduced into the lattice increased. For example, at a treatment temperature of 160 °C, the [Mn]/[Mn + Cd] ratio introduced during synthesis was identical to the parameter x obtained in the final compound (x = 0.4, 0.6, and 0.8).

Appendix A shows X-ray patterns for the Mn0.6 HT120 photocatalyst with deposited nickel sulfide (0.1–1.0 wt.%). For the NiS/Mn0.6 HT120 samples, no characteristic peaks of the NiS particles were observed, likely due to the high dispersity (according to micrographs) and low crystallinity of the deposited NiS nanoparticles. The parameter x for Cd_1−x_Mn_x_S and the volume of the unit cell did not change when the surface of the photocatalysts was modified with NiS, which indicates the presence of nickel sulfide only on the surface of the Cd_1−x_Mn_x_S solid solution. The phase composition of all the samples is presented in Table 1.

#### 3.1.2. UV-Vis Diffuse Reflectance Spectroscopy Analysis

Figure 3 shows the diffuse reflectance spectra of the samples of the HT120 and HT140 series. These samples are characterized by high absorption in the visible range. The optical band gap was estimated using Tauc’s coordinates for a direct allowed transition. An accurate calculation of the band gap is difficult due to the complex shape of the spectra due to the presence of impurities. It was shown that the band gap energy grows with the x parameter in Cd_1−x_Mn_x_S. According to the XRD data, for the Mn0.4 HT120 photocatalyst, the formation of a pure solid solution was observed and the band gap of this sample was ca. 2.40 eV, whereas for Mn0.6 HT120, an increase in the band gap to ca. 2.5 eV due to the increase of Mn content was observed. The same tendency was observed for Mn0.4 HT140 and Mn0.6 HT140 photocatalysts.

Comparison of the diffuse reflectance spectra of the pristine and nickel-modified samples (Appendix A) indicates an increase in the absorption of visible light in the region of 450–800 nm. This dependence correlates with an increase in the mass fraction of deposited NiS, associated with the absorption of the deposited nickel sulfide particles in the region of 300–800 nm [35].

#### 3.1.3. HRTEM and Element Mapping Analysis

HRTEM images with element mapping were obtained for the Mn0.8 HT140 sample (Figure 4). The element mapping confirmed the presence of the Cd_1−x_Mn_x_S solid solution in the sample. One can see that oxide impurities were distributed relatively evenly over the surface, in close contact with sulfide particles. The localization of oxygen-containing agglomerates at the boundaries of solid solution particles was observed. An analysis of the interplanar distance (Figure 4f) showed the presence of the Cd_1−x_Mn_x_S solid solution, manganese oxide, and manganese hydroxide. Thus, the interplanar spacing shown in Figure 4f of approximately 0.31 nm corresponds to the interplanar spacing d_002_ in the structure of the Cd_1−x_Mn_x_S solid solution [23]. The spherical particles seen in Figure 4e are also characteristic of Cd_1−x_Mn_x_S. The interplanar spacing equal to 0.49 nm corresponds to the d_101_ plane of manganese oxide β-Mn_3_O_4_ [36] (PDF no. 24-734), while an interplanar spacing equal to 0.38 nm is typical for Mn(OH)_2_ (PDF no. 41-1379).

The HRTEM images of 0.1% NiS/Mn0.6 HT120 are represented in Figure 5 and SEM images of this sample are represented in Appendix A. SEM images show a loose porous structure forming aggregates up to several micrometers in size. It can be seen from HRTEM images that NiS nanoparticles with a particle size of less than 2 nm covered the surface of Cd_1−x_Mn_x_S. Interplanar spaces of 0.34 and 0.29 nm corresponding to the d_002_ plane of Cd_1−x_Mn_x_S and the d_102_ plane of NiS, respectively, were observed. This confirms the formation of a p–n heterostructure at the interface between NiS and the Cd_1−x_Mn_x_S solid solution.

Moreover, according to the TEM element mapping data, particles of nickel were deposited uniformly on the surface of the solid solution. Based on the XRD results, this sample contained manganese oxide and hydroxide, which is consistent with the interplanar distances found in Figure 5f.

### 3.2. Photocatalytic Activity

At the beginning, we studied the dependence of the rate of photocatalytic hydrogen evolution on the x parameter in Cd_1−x_Mn_x_S and the temperature of hydrothermal treatment. A standard Na_2_S/Na_2_SO_3_ mixture (λ = 425 nm) was used as an electron donor. The sulfide/sulfite system is not only considered as an electron donor but also protects the sulfide photocatalyst from anodic photocorrosion. Sulfide in CdS structure can be easily oxidized by photogenerated holes, and Cd^2+^ ions from the photocatalyst leach into the aqueous solution. When sodium sulfide is used as a sacrificial agent, dissolved sulfide anions are oxidized instead of lattice sulfide anions in the CdS structure. The results obtained are presented in Table 2 and Figure 6.

Based on the data, the pristine solid solution (e.g., sample Mn0.4 HT160 (Cd_0.62_Mn_0.38_S)) or the multiphase samples Mn0.6 HT120 (Cd_0.54_Mn_0.46_S, Mn(OH)_2_), and Mn0.6 HT140 (Cd_0.53_Mn_0.47_S, β-Mn_3_O_4_, and Mn(OH)_2_) possessed high photocatalytic activity. In the case of the multiphase samples, the formation of effective heterojunctions required the presence of the Cd_1−x_Mn_x_S solid solution, manganese oxide, and manganese hydroxide or two components—the Cd_1−x_Mn_x_S solid solution and manganese hydroxide. In our previous research, we constructed a scheme of heterojunctions among Cd_1−x_Mn_x_S, manganese oxide, and manganese hydroxide [34]. One can notice a general trend, with a greater introduction of manganese cations into the structure of the solid solution, resulting in a noticeable increase in activity. The solid solution has a higher activity due to the more negative position of the electrochemical level of the conduction band and, therefore, higher reduction ability of photogenerated electrons compared with the pristine cadmium sulfide. However, this material still has good sensitivity to visible light [34].

The formation of MnS particles (Table 3) led to a decrease in the activity of the photocatalysts due to the large band gap of manganese sulfide (3.54 eV), which prevented the absorption of visible radiation. Accordingly, in this case, the formation of heterojunctions for the Mn0.8 HT120 sample, which consisted of Cd_0.63_Mn_0.37_S, β-Mn_3_O_4_, and MnS, did not appear to be effective. Finally, for the two-phase Cd_1−x_Mn_x_S-β-Mn_3_O_4_ samples (Mn0.6 HT100/160, and Mn0.4 HT 100/140), relatively low reaction rates were observed, associated with the low efficiency of this heterojunction for separating photogenerated electrons and holes.

On the basis of the analysis of the synthesized photocatalysts, a sample of Mn0.6 HT120 (Cd_0.54_Mn_0.46_S, Mn(OH)_2_) was chosen for further modification by NiS deposition for improving the photocatalytic activity in the target process. In the case of modified photocatalysts, we obtained hydrogen not only from the Na_2_S/Na_2_SO_3_ aqueous solution but also from the α-D glucose basic aqueous solution. In the case of the use of sugars as electron donors, hydrogen evolution can be accompanied by the production of valuable organic compounds, such as acetic and formic acid. At the same time, the use of a system consisting of biomass components is attractive from a practical point of view. In this case, it is possible to obtain “green” hydrogen using only water, solar irradiation, and plant biomass components. According to the literature data, the process of photocatalytic hydrogen evolution from sugar solutions by adding samples based on Cd_1−x_Mn_x_S has not been studied before [37].

When nickel sulfide was deposited on the surface of a solid solution (0.5% NiS/Mn0.6 HT120), an increase in the rate of photocatalytic hydrogen evolution was observed from 10.8 to 34.2 mmol g^−1^ h^−1^ in the case of the use of Na_2_S/ Na_2_SO_3_ as an electron donor (Figure 7). For the glucose aqueous solution, the hydrogen evolution rate grew from 0.2 to 5.8 mmol g^−1^ h^−1^ after the deposition of NiS over the surface of Mn0.6 HT120 (Figure 7). According to the XRD data, the sample Mn0.6 HT120 consists mostly of Cd_0.54_Mn_0.46_S solid solution. Based on data of TEM images, the formation of heterojunctions occurred with close contact of the phases. According to literature, the position of the conduction band for Cd_0.5_Mn_0.5_S vs. NHE was –0.8, and the VB position was 1.49 eV [23].
(3)$$\mathrm{NiS}+e^{-}+H^{+}\to \mathrm{HNiS}$$
(4)$$\mathrm{HNiS}+e^{-}+H^{+}\to H_{2}+\mathrm{NiS}$$

Based on the presented data, photoexcited electrons can transfer from the VB of the Cd_0.54_Mn_0.46_S to the CB; then, photoexcited electrons migrate to the NiS to form the HNiS intermediate (Equation (3)), which participate in the reduction of protons to H_2_ (Equation (4)) [23]. It was assumed that NiS plays a similar role as most of the noble metals because it is known that NiS usually couples properties of a metal and a semiconductor, owing to its very small band gap [24]. At the same time, holes oxidize the sacrificial agents, preventing the recombination of electron–hole pairs. The scheme of separation of charge carriers is shown in Figure 7b.

When the mass content of NiS reached 1 wt.%, the reaction rate decreased, likely due to an increase in the scattering of incident light by the NiS particles. As well as a decrease in the absorption of incident irradiation by the catalyst, an excessive amount of NiS will lead to an excess content of Ni^2+^ cations, which will act as a recombination center for charge carriers [38].

Noble metals, as previously known, are effective co-catalysts for hydrogen evolution due to the high value of the electron work function (in the case of platinum, the electron work function is 5.4 eV) [39]. Thus, we prepared the photocatalyst 1 wt.% Pt/ Mn0.6 HT120 for comparison with non-noble metal photocatalysts. It was shown that the deposition of platinum on the surface of the photocatalyst did not lead to an increase in activity when studying the process of hydrogen evolution from the aqueous solution of Na_2_S/Na_2_SO_3_ (Figure 7). A noticeable increase in activity from 0.2 to 5.8 mmol g^−1^ h^−1^ was observed in the glucose solution due to the peculiarity of the platinum particles deposited on the surface, which promote the adsorption of organic substrates [40]. The data on the photocatalytic activities of the best synthesized photocatalysts are summarized in Table 3.

Based on the data presented in Table 3, the highest rate of hydrogen production from the Na_2_S/Na_2_SO_3_ aqueous solutions was 34.2 mmol g^−1^ h^−1^, and the apparent quantum efficiency was 15.4% for the 0.5% NiS/Mn0.6 photocatalyst. When using glucose as a substrate for the same sample, these values were equal to 5.8 mmol g^−1^ h^−1^ and 2.9%, respectively. The highest activity in the case of the use of glucose as an electron donor, 7.3 mmol g^−1^ h^−1^, was possessed by the 1 wt.% Pt/ Mn0.6 HT120 photocatalyst. It should be noted that in the case of using glucose solutions, the activity of the photocatalyst with the deposited nickel sulfide, 0.5% NiS/Mn0.6 HT120, was only 21% lower than the activity of the platinized sample, 1% Pt/Mn0.6 HT120. It can be concluded that the modification of photocatalysts with platinum does not lead to a fundamental difference in activity, although it requires expensive reagents.

A comparison of the obtained numerical values with the literature data is presented in Table 4. The highest values of the quantum efficiency of the synthesized samples are at the level of [22] or are exceeded several times by [23] previously published works for the sulfide system of electron donors. However, in this work, we used sodium sulfide as a precursor of the sulfur-containing component, which is economically advantageous. In the case of a glucose solution system, data on the use of photocatalysts based on Cd_1−x_Mn_x_S in the process of hydrogen evolution have not been published before. Previous authors [41] investigated the target process with the addition of 0.5 wt.% PtO_x_/Cd_0.7_Zn_0.3_S/ZnS, and the rate of hydrogen evolution was 3.4 mmol g^−1^ h^−1^, which was lower than that obtained in this work using photocatalysts modified with nickel sulfide. 

### 3.3. Stability Tests

In addition to activity, the decisive factor is the stability of photocatalysts. To study the stability of the photocatalysts, long-term experiments on the evolution of hydrogen from organic and inorganic electron donors were carried out. The duration of one cycle was 1–1.5 h. After each cycle, the reactor was purged with argon for 20 min. The following samples were selected for this study: 0.5% NiS/Mn0.6 HT120, 1% NiS/Mn0.6 HT120, and 1% Pt/Mn0.6 HT120.

First, stability tests were carried out in the Na_2_S/Na_2_SO_3_ aqueous solution (Figure 8). Previously, it was reported that photocorrosion of Cd_1−x_Mn_x_S solid solutions can occur in a solution of sodium sulfide and sodium sulfite [43]. As shown in Figure 8, when 1% NiS/Mn0.6 HT120 was added, after one cycle, a decrease in activity by approximately 20% was observed, after which the reaction rate did not change. This decrease in activity is likely due to the enlargement of nickel-containing particles under the reaction conditions, which follows from the XPS data on a decrease in the [Ni]/([Mn] + [Cd]) ratio (Table 5). At the same time, the surface content of NiS_x_ did not change, which likely stabilized the activity of the photocatalyst.

Figure 9 shows the Ni*2p_3/2_* spectra of the nickel-containing samples. The Ni*2p_3/2_* spectrum of a freshly prepared sample contained two peaks at 853.0 and 855.5 eV. The first peak corresponds to nickel in the composition of nickel sulfide [44,45], while the position of the second peak corresponds to nickel in the Ni^2+^ state, likely in the structure of nickel sulfite/sulfite or nickel hydroxide [46]. After the reaction, the spectrum of Ni*2p_3/2_* also showed two peaks in the region corresponding to nickel in nickel sulfide and in the oxidized state.

Thus, when a photocatalytic reaction was carried out in Na_2_S/Na_2_SO_3_ solution, there was practically no deactivation of the photocatalyst. The 1% Pt/Mn0.6 HT120 photocatalyst did not show an increase in activity compared with the pure solid solution in the reaction of hydrogen evolution from the Na_2_S/Na_2_SO_3_ solution. Therefore, this sample was not studied in the long-term experiments.

The stability of 0.5% NiS/Mn0.6 HT120 and 1% Pt/Mn0.6 HT120 was tested in a photocatalytic hydrogen evolution from the glucose aqueous solution. In the case of the photocatalyst modified with nickel sulfide, the reaction rate increased after one cycle and then remained practically unchanged. As in the case of the sulfide system of the electron donors, the nickel-containing particles slightly coarsened under the reaction conditions and then remained unchanged, so no significant decrease in activity was observed. In contrast, for the platinized photocatalyst, a noticeable deactivation was observed.

Figure 10 shows the Pt*4f* spectra of the fresh samples and those using 1% Pt/Mn0.6 HT120. For the Pt*4f_7/2_*–Pt*4f_5/2_* platinum doublet, spin–orbit splitting was 3.33 eV [47]. The spectra were approximated by one Pt*4f_7/2_*–Pt*4f_5/2_* doublet with a Pt*4f_7/2_* binding energy in the region of 72.6 eV. Thus, we can conclude that platinum is involved in the oxidized Pt^2+^ state on the surface of photocatalysts. Likely, during the course of the reaction, Pt particles significantly coarsen and photocorrosion of the catalyst occurs, which follows from a decrease in the ratio [Pt]/([Pt] + [Cd]), based on the data presented in Table 5. These factors are the reason for the deactivation of the 1% Pt/Mn0.6 HT120 photocatalyst.

Figure 11 shows that the activity of the 0.5% NiS/Mn0.6 HT120 photocatalyst in the fourth run was even higher than the activity of the platinized photocatalyst under the same conditions. Thus, we can conclude that systems based on NiS/Cd_1−x_Mn_x_S are promising due to their high stability both in Na_2_S/Na_2_SO_3_ solutions and in glucose solutions. On the contrary, when performing photocatalytic experiments in the presence of platinum-containing compounds in a glucose solution, the photocatalyst is deactivated due to the enlargement of Pt particles and photocorrosion of the sample.

## 4. Conclusions

Herein, we proposed new materials based on Cd_1−x_Mn_x_S that are effective in the process of photocatalytic hydrogen evolution under visible light with a wavelength of 425 nm. When using the Na_2_S/Na_2_SO_3_ system as an electron donor, the maximum rate of hydrogen evolution reached the superior level 10.8 and 13.7 mmol g^–1^ h^−1^ for the photocatalysts Mn0.6 HT120 (Cd_0.54_Mn_0.46_S/Mn(OH)_2_) and Mn0.4 HT160 (Cd_0.62_Mn_0.38_S), respectively. The high activity observed is likely due to the formation of a solid solution with an energy structure suitable for hydrogen evolution and the occurrence of interfacial heterojunctions. The deposition of nickel sulfides led to a significant increase in the rate of photocatalytic hydrogen evolution from the Na_2_S/Na_2_SO_3_ aqueous solution by a factor of three and amounts to the extraordinary value 34.2 mmol g^–1^ h^−1^ (AQE = 15.4%) for a sample of 0.5% NiS/Mn0.6 HT120. This increase in activity is associated with the occurrence of interphase heterojunctions between NiS and Cd_1−x_Mn_x_S.

When studying the process of hydrogen evolution from a glucose solution, the rate of H_2_ evolution for a sample of 0.5% NiS/Mn0.6 HT120 was 5.8 mmol g^–1^ h^−1^, while for the platinized photocatalyst 1% Pt/Mn 0.6 HT120, the activity was only 27% higher and achieved 7.3 mmol g^–1^ h^−1^. The stability of the photocatalysts was studied in cyclic experiments. It was shown that the NiS/Mn0.6 HT120 samples were not subjected to deactivation, either from an equimolar Na_2_S/Na_2_SO_3_ mixture or from a glucose solution. In contrast, for the 1% Pt/Mn0.6 HT120 sample, a noticeable deactivation occurred, likely associated with the coarsening of platinum particles in the course of the reaction and photocorrosion of the material. Thus, it can be concluded that the hydrothermal synthesis of Cd_1−x_Mn_x_S with further modification of the surface with nickel sulfide particles can be considered in the future as a replacement for expensive noble metals.

A distinctive feature of this study is the comparison of organic and inorganic electron donors, namely Na_2_S/Na_2_SO_3_ and glucose, in photocatalytic hydrogen production. According to the literature data, the process of photocatalytic hydrogen evolution from sugar solutions by adding samples based on Cd_1−x_Mn_x_S has not previously been studied. At the same time, the use of a system consisting of biomass components is attractive not only from a practical point of view but also from an economic one. In this case, it is possible to obtain “green” hydrogen using only water, solar irradiation, and plant biomass components.

## Figures and Tables

**Figure 1 materials-15-08026-f001:**
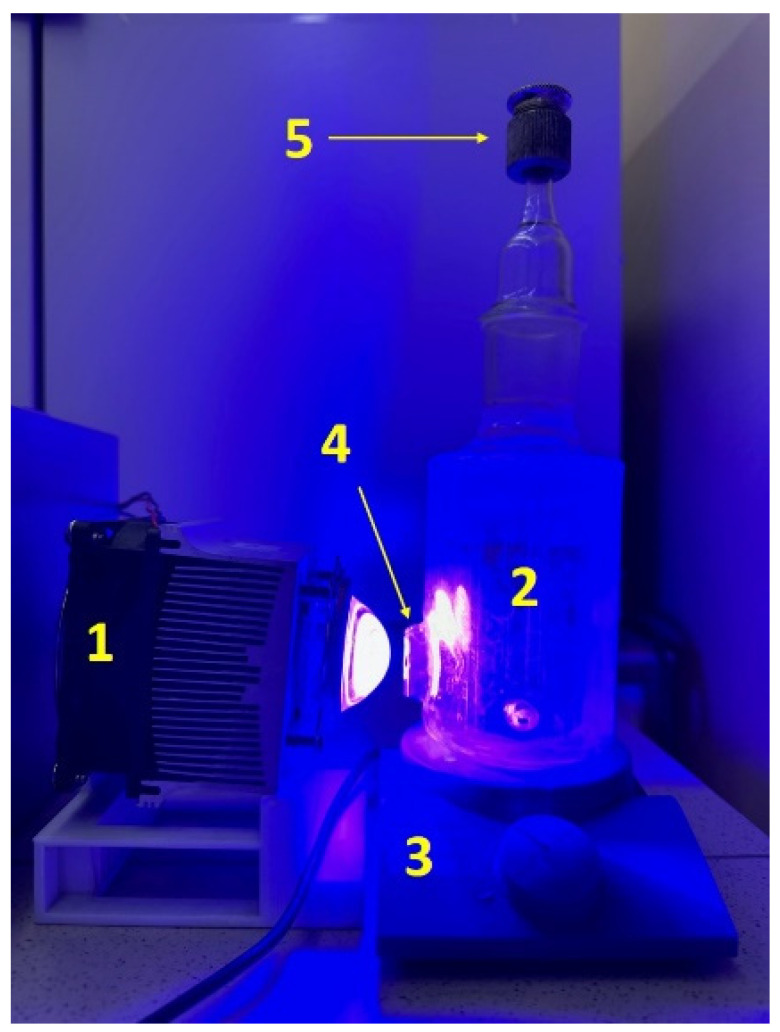
Photocatalytic setup for hydrogen evolution: 1—LED (425 nm); 2—suspension; 3—magnetic stirrer; 4—a quartz window; and 5—sampler.

**Figure 2 materials-15-08026-f002:**
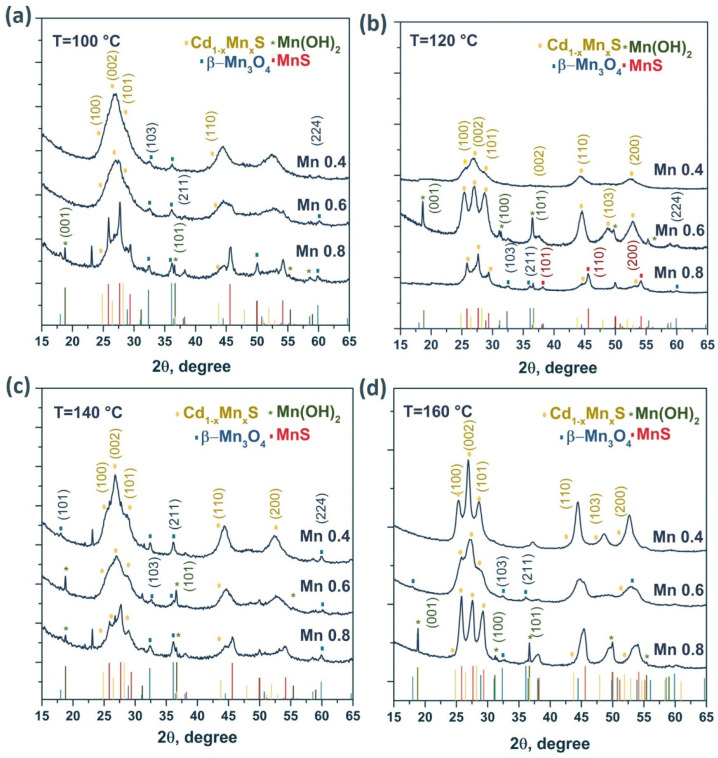
XRD patterns of the samples Cd_1−x_Mn_x_S with different manganese content for the HT100—HT160 series (**a**–**d**).

**Figure 3 materials-15-08026-f003:**
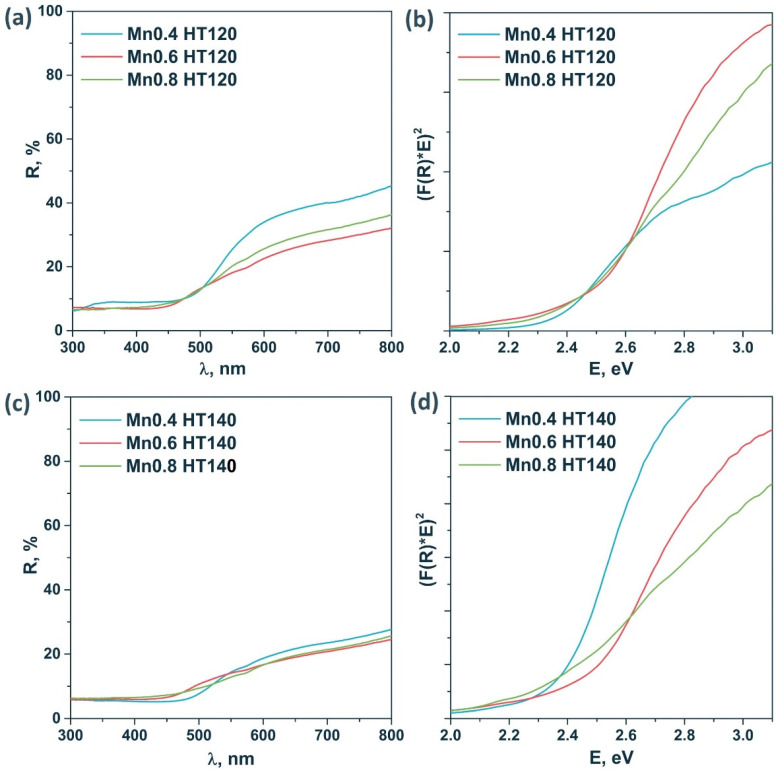
Diffuse reflectance spectra (**a**,**c**) and Tauc’s plot (**b**,**d**) of the sample series Cd_1−x_Mn_x_S HT120 and Cd_1−x_Mn_x_S HT140.

**Figure 4 materials-15-08026-f004:**
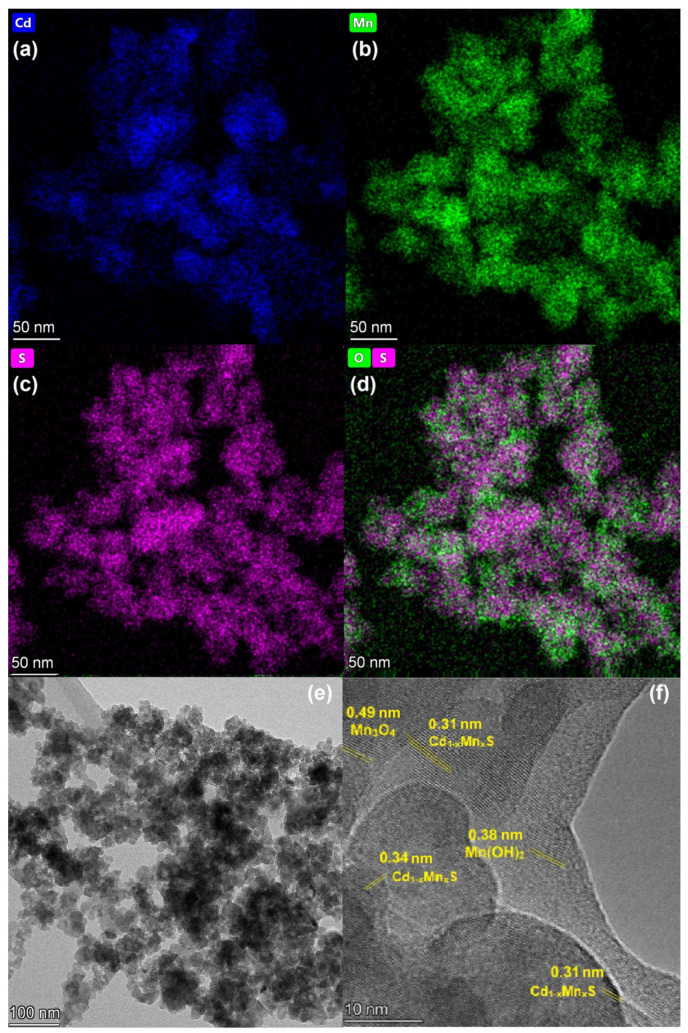
Element mapping (**a**–**d**) and TEM images (**e**,**f**) of Mn0.8 HT140 photocatalyst.

**Figure 5 materials-15-08026-f005:**
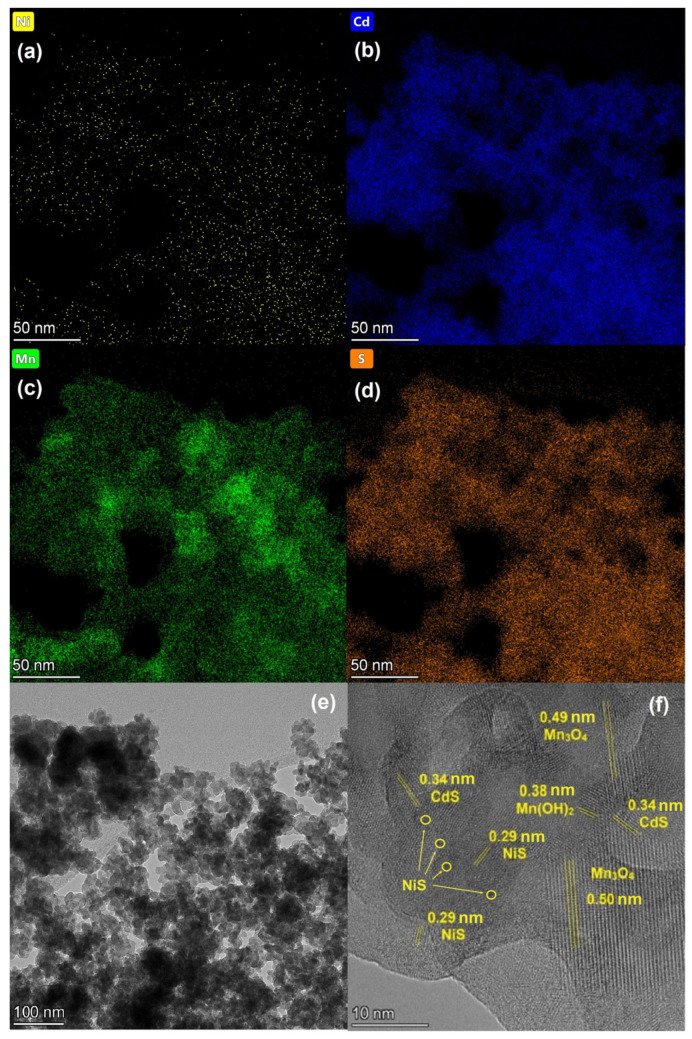
Element mapping (**a**–**d**) and TEM images (**e**,**f**) of 0.1% NiS/Mn0.6 HT120 photocatalyst.

**Figure 6 materials-15-08026-f006:**
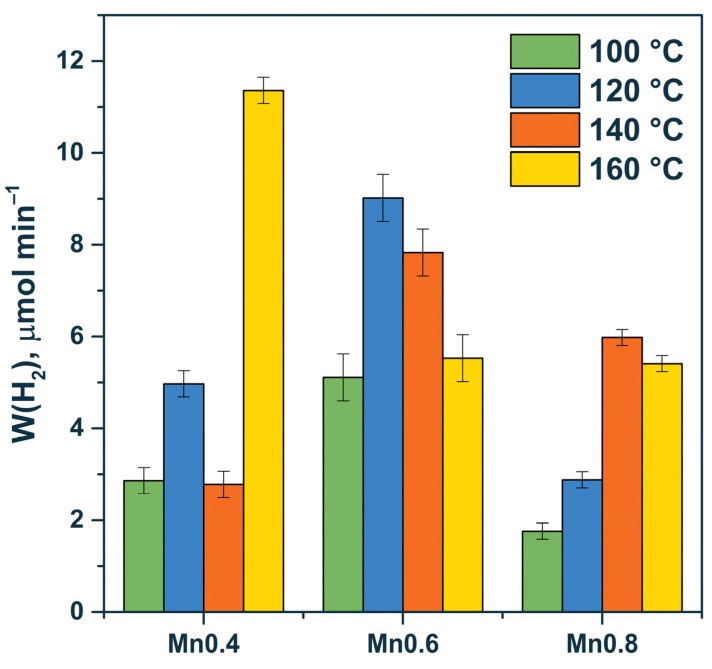
Dependence of the rate of photocatalytic hydrogen evolution on the manganese content for HT100-HT160 series. Experimental conditions: C_0_(Na_2_S/Na_2_SO_3_) = 0.1/0.1 M, m_cat_ = 50 mg, V_susp_ = 100 mL, and λ = 425 nm.

**Figure 7 materials-15-08026-f007:**
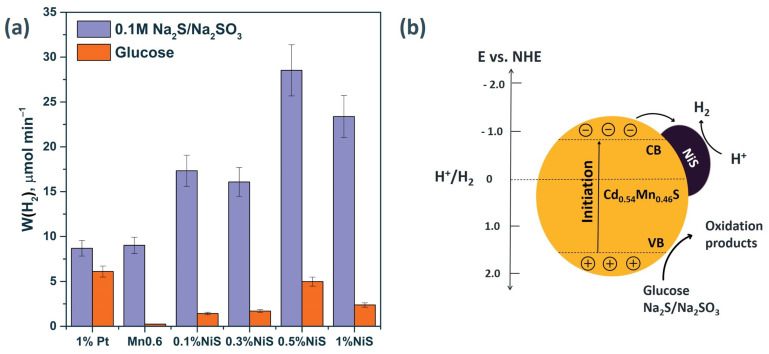
(**a**)The rate of photocatalytic hydrogen evolution from Na_2_S/Na_2_SO_3_ and glucose aqueous solutions. Experimental conditions: C_0_(Na_2_S/Na_2_SO_3_) = 0.1/0.1 M, C_0_(glucose) = 0.012 M, C_0_(NaOH) = 0.1 M m_cat_ = 50 mg, V_susp_ = 100 mL, and λ = 425 nm. (**b**) Simplified charge separation scheme for NiS/Cd_0.54_Mn_0.46_S.

**Figure 8 materials-15-08026-f008:**
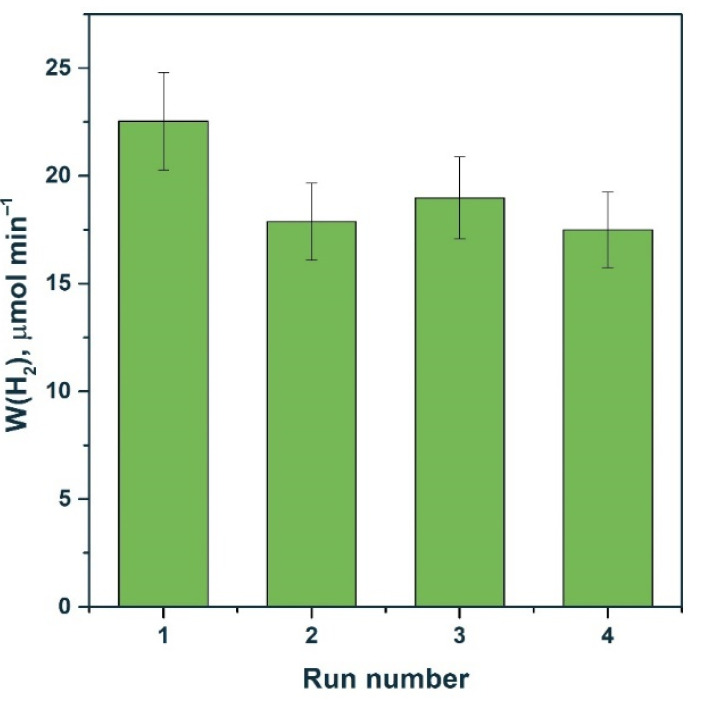
Stability test of 1% NiS/Mn0.6 HT120 photocatalyst. Experimental conditions: C_0_(Na_2_S/Na_2_SO_3_) = 0.1/0.1 M, m_cat_ = 50 mg, V_susp_ = 100 mL, and λ = 425 nm.

**Figure 9 materials-15-08026-f009:**
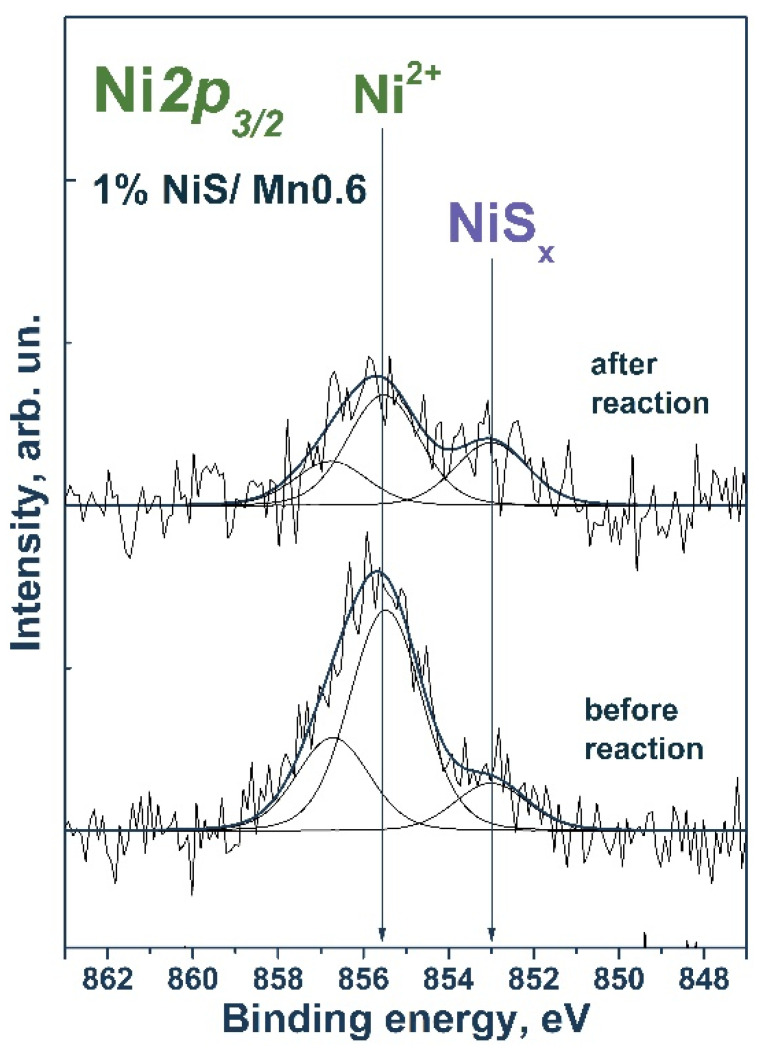
Ni*2p_3/2_* spectrum of the studied samples. The spectrum is normalized to the integral total intensity of the corresponding Mn*2p* spectra.

**Figure 10 materials-15-08026-f010:**
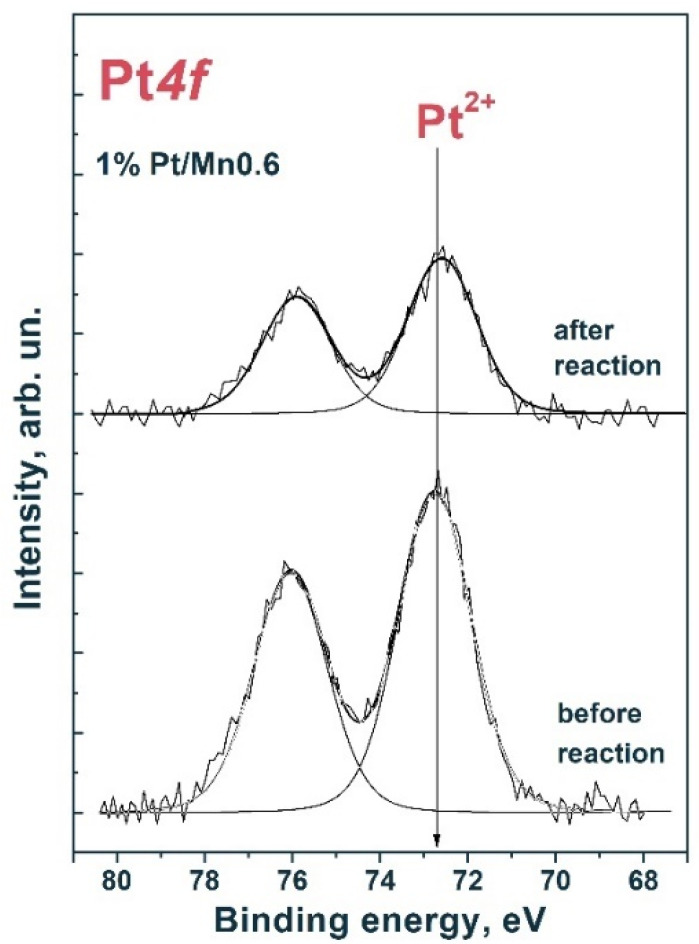
Pt*4f* spectrum of fresh samples and those using 1% Pt/Mn0.6 HT120. The spectrum is normalized to the integral total intensity of the corresponding Cd*3d* and Mn*2p* spectra.

**Figure 11 materials-15-08026-f011:**
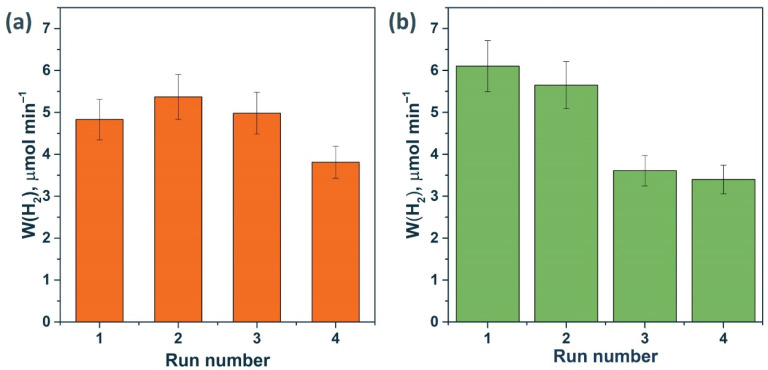
Stability test of photocatalysts (**a**) 0.5% NiS/Mn0.6 HT120 and (**b**) 1% Pt/Mn0.6 HT120. Experimental conditions: C_0_(glucose) = 0.012 M, C_0_(NaOH) = 0.1 M, m_cat_ = 50 mg, V_susp_ = 100 mL, λ = 425 nm, and t_run_ = 90 min.

**Table 1 materials-15-08026-t001:** Phase composition and lattice parameters of all synthesized samples.

Sample	Cd_1−x_Mn_x_S			β-Mn_3_O_4_, nm	Mn(OH)_2_, nm	MnS, nm	Phase Composition
	D, nm	V, Å^3^	x				
Cd_1−x_Mn_x_S HT100 series							
Mn0.4	2.3	95.5	0.38	8	-	-	Cd_0.62_Mn_0.38_S, β-Mn_3_O_4_
Mn0.6	1.9	93.5	0.55	9	-	-	Cd_0.45_Mn_0.55_S, β-Mn_3_O_4_
Mn0.8	1.9	92.8	0.61	10	>100	22	Cd_0.39_Mn_0.61_S, β-Mn_3_O_4_, Mn(OH)_2_, MnS
Cd_1−x_Mn_x_S HT120 series							
Mn0.4	6.9	97.8	0.17	-	-	-	Cd_0.83_Mn_0.17_S
Mn0.6	5.1	94.6	0.46	-	35	-	Cd_0.54_Mn_0.46_S, Mn(OH)_2_
Mn0.8	4.2	95.6	0.37	6.5	-	11	Cd_0.63_Mn_0.37_S, β-Mn_3_O_4_, MnS
Cd_1−x_Mn_x_S HT140 series							
Mn0.4	3.0	96.1	0.32	10	-	-	Cd_0.68_Mn_0.32_S, β-Mn_3_O_4_
Mn0.6	2.6	94.4	0.47	10	>100	-	Cd_0.53_Mn_0.47_S, β-Mn_3_O_4_, Mn(OH)_2_
Mn0.8	2.4	93.2	0.58	11	>100	12	Cd_0.42_Mn_0.58_S, β-Mn_3_O_4_, Mn(OH)_2_, MnS
Cd_1−x_Mn_x_S HT160 series							
Mn0.4	6.4	95.4	0.38	-	-	-	Cd_0.62_Mn_0.38_S
Mn0.6	3.3	92.6	0.63	5.1	-	-	Cd_0.37_Mn_0.63_S, β-Mn_3_O_4_
Mn0.8	7.2	90.2	0.84	-	>100	-	Cd_0.16_Mn_0.84_S, Mn(OH)_2_

**Table 2 materials-15-08026-t002:** The rate of hydrogen evolution in dependence on the parameter x in Cd_1−x_Mn_x_S and the temperature of hydrothermal treatment. Experimental conditions: C_0_(Na_2_S/Na_2_SO_3_) = 0.1/0.1 M, m_cat_ = 50 mg, V_susp_ = 100 mL, and λ = 425 nm.

X in Cd_1−x_Mn_x_S	W, mmol g^−1^ h^−1^ Hydrothermal Treatment Temperature			
	T = 100 °C	T = 120 °C	T = 140 °C	T = 160 °C
0.4	3.5	3.5	3.4	13.7
0.6	6.1	10.8	9.4	6.6
0.8	2.2	6.0	7.2	6.5

**Table 3 materials-15-08026-t003:** The rate of photocatalytic hydrogen evolution for different photocatalysts. Experimental conditions: C_0_(Na_2_S/Na_2_SO_3_) = 0.1/0.1 M, m_cat_ = 50 mg, V_susp_ = 100 mL, and λ = 425 nm.

Sample	W(H_2_), mmol g^−1^ h^−1^		AQE, %	
	Na_2_S/Na_2_SO_3_	Glucose	Na_2_S/Na_2_SO_3_	Glucose
Mn0.6 HT120	10.8	0.2	5.4	0.1
0.1% NiS/Mn0.6 HT120	20.8	1.7	10.4	0.8
0.3% NiS/Mn0.6 HT120	19.3	2.0	9.6	1.0
0.5% NiS/Mn0.6 HT120	34.2	5.8	15.4	2.9
1% NiS/Mn0.6 HT120	28.1	2.9	14.0	1.4
1% Pt/Mn0.6 HT120	10.4	7.3	5.2	3.6

**Table 4 materials-15-08026-t004:** The comparison of the activities of the synthesized samples with recently published data.

Sample	Synthesis	Light Source	Cut-Off Filter	Electron Donor	W, mmol g^−1^ h^−1^	AQE, %	Ref.
Cd_0.7_Mn_0.3_S	Hydrothermal synthesis; thioacetamide. Hydrothermal synthesis; Ni(CH_3_COO)_2_; EDTA.	300 W Xe lamp	λ ≥ 420 nm	Na_2_S/Na_2_SO_3_	20		[22]
0.5 wt.% NiS/Cd_0.7_Mn_0.3_S					42		
1 wt.% NiS/ Cd_0.7_Mn_0.3_S					66	20.2	
3 wt.% NiS/ Cd_0.7_Mn_0.3_S					38		
5 wt.% NiS/ Cd_0.7_Mn_0.3_S					35		
1 wt.% Pt/ Cd_0.7_Mn_0.3_S					43		
Cd_0.5_Mn_0.5_S	Hydrothermal synthesis; thioacetamide. Cation exchange; Ni(NO_3_)_2_; Na_2_S.	300 W Xe lamp	λ ≥ 420 nm	Na_2_S/Na_2_SO_3_	0.6		[23]
0.3 wt.% NiS/ Cd_0.5_Mn_0.5_S					0.84	5.2	
CdS	Hydrothermal synthesis; thiourea. Hydrothermal synthesis; Ni(CH_3_COO)_2_; thiourea.	300 W Xe lamp	λ ≥ 420 nm	0.35M Na_2_S/ 0.25M Na_2_SO_3_	0.05		[42]
5 wt.% NiS/CdS					1.1	6.1	
0.5 wt.% PtO_x_/Cd_0.7_Zn_0.3_S/ZnS	Codeposition method; CdCl_2_; Zn(NO_3_)_2_; Na_2_S. Soft chemical reduction; H_2_PtCl_2_; NaBH_4_.	450 nm LED	λ ≥ 450 nm	α-D glucose/ NaOH	3.4	-	[41]
Cd_0.4_Mn_0.6_S	Hydrothermal synthesis; Cation exchange; Ni(NO_3_)_2_; Na_2_S.	450 nm LED	λ ≥ 425 nm	Na_2_S/ Na_2_SO_3_	10.8	5.4	Current study
0.5 wt.% NiS/ Cd_0.4_Mn_0.6_S					34.2	15.4	
				α-D glucose/ NaOH	4.8	2.9	

**Table 5 materials-15-08026-t005:** Atomic ratios of elements in the near-surface layer of samples according to XPS data.

No.	Sample	[M]/([M] + [Cd])	[Mn]/([Mn] + [Cd])	[S]/([Mn] + [Cd])	[S^2−^]/([MnS_x_] + [CdS_x_])	%, MnS_x_
Na_2_S/Na_2_SO_3_						
1	1% NiS/Mn0.6 HT120	0.0073	0.158	0.99	0.95	59
2	1% NiS/Mn0.6 HT120 *	0.0041	0.125	1.09	0.99	52
Glucose						
3	1% Pt/Mn0.6 HT120	0.0198	0.406	1.47	1.58	54
4	1% Pt/Mn0.6 HT120 *	0.0091	0.430	1.03	1.17	55

* after photocatalytic tests; M = Ni or Pt.

## Data Availability

The data presented in this study are available on request from the corresponding author.

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
