# Peer review of "Efficient Photocatalytic Hydrogen Production over NiS-Modified Cadmium and Manganese Sulfide Solid Solutions"

_materials, 2022, doi:10.3390/ma15228026_

Round 1

Reviewer 1 Report

In the present work, an efficient photocatalytic effect of a new catalyst based on NiS-modified cadmium and manganese sulfide for hydrogen production has been investigated. Characterization was performed using various methods, including XRD, XPS, HR TEM, and UV-vis spectroscopy. The photocatalytic activity was tested in hydrogen evolution from aqueous solutions of Na2S/Na2SO3 and glucose under visible light (425 nm). The best activity in the case of the modified samples was shown by 0.5 wt.% NiS/Cd0.4Mn0.6S (T = 120 ºC) at the level of 34.2 mmol g–1h–1 (AQE 14.4%) for the Na2S/Na2SO3 solution and 4.6 mmol g–1h–1 (AQE 2.9%) for the glucose solution. The nickel-containing samples possessed high stability, in solutions of both sodium sulfide/sulfite and glucose. Thus, nickel sulfide is considered an alternative to depositing precious metals, which is attractive from an economic point of view. In conclusion: The manuscript is suitable for publication in Materials after minor revision

 Some important points should be addressed while submitting the revised form of this work:

1. The introduction part seems to be interesting. Unfortunately, many old references have been cited. The authors are highly appreciated to add up-to-date references (2022) to the work.

2. The findings/novelty of the present work should be clearly highlighted in the revised text.

3. The presented XRD analyses are quite good. The authors can give a more detailed study of the XRD results. They can calculate the lattice parameters of the studied samples, such as crystallite size, lattice strain, and dislocation density of your material, with full discussions. Please also add hkl on the XRD charts. This will give more enhancement efforts to the study.

4. The produced materials required some additional investigation characteristics such as PET, FI-SEM, and AFM.

5. A simple illustration scheme with explanation text for the photocatalytic process (mechanism of action) is highly appreciated to be added to the revised text.

6. The graphical abstract is also appreciated for the present work, which will give the reader a good feeling about the manuscript content, therefore the authors are highly appreciated for designing a creative interesting graphical abstract for this work. Or by other words, the authors can add it as an illustration scheme inside the discussion part.

7. Few mistakes are found in the text, the authors are kindly required to recheck the whole revised text for any typos and grammatical errors. 

Author Response

 Some important points should be addressed while submitting the revised form of this work:

  1. The introduction part seems to be interesting. Unfortunately, many old references have been cited. The authors are highly appreciated to add up-to-date references (2022) to the work.

Reply: We have replaced links to earlier work with articles published in 2022.

New references:

  1. Vostakola, F.; Salamatinia, M.; Horri, A.; Fallah Vostakola, M.; Salamatinia, B.; Horri, B.A. A Review on Recent Progress in the Integrated Green Hydrogen Production Processes. Energies 2022, Vol. 15, Page 1209 2022, 15, 1209, doi:10.3390/EN15031209.
  2. Chen, W.H.; Lee, J.E.; Jang, S.H.; Lam, S.S.; Rhee, G.H.; Jeon, K.J.; Hussain, M.; Park, Y.K. A Review on the Visible Light Active Modified Photocatalysts for Water Splitting for Hydrogen Production. J. Energy Res. 2022, 46, 5467–5477, doi:10.1002/ER.7552.
  3. Estévez, R.A.; Espinoza, V.; Ponce Oliva, R.D.; Vásquez-Lavín, F.; Gelcich, S. Multi-Criteria Decision Analysis for Renewable Energies: Research Trends, Gaps and the Challenge of Improving Participation. 2021, Vol. 13, Page 3515 2021, 13, 3515, doi:10.3390/SU13063515.
  4. Feng, C.; Wu, Z.P.; Huang, K.W.; Ye, J.; Zhang, H. Surface Modification of 2D Photocatalysts for Solar Energy Conversion. Mater. 2022, 34, 2200180, doi:10.1002/ADMA.202200180.
  5. Pandey, P.; Ingole, P.P.; Pandey, P.; Ingole, Á.P.P. Emerging Photocatalysts for Hydrogen Production. 2022, 647–671, doi:10.1007/978-3-030-77371-7_21.
  6. Sahani, S.; Malika Tripathi, K.; Il Lee, T.; Dubal, D.P.; Wong, C.P.; Chandra Sharma, Y.; Young Kim, T. Recent Advances in Photocatalytic Carbon-Based Materials for Enhanced Water Splitting under Visible-Light Irradiation. Energy Convers. Manag. 2022, 252, 115133, doi:10.1016/J.ENCONMAN.2021.115133.
  7. Prusty, D.; Paramanik, L.; Parida, K. Recent Advances on Alloyed Quantum Dots for Photocatalytic Hydrogen Evolution: A Mini-Review. Energy and Fuels 2021, 35, 4670–4686, doi:10.1021/acs.energyfuels.0c04163.
  8. Wang, Y.Y.; Chen, Y.X.; Barakat, T.; Zeng, Y.J.; Liu, J.; Siffert, S.; Su, B.L. Recent Advances in Non-Metal Doped Titania for Solar-Driven Photocatalytic/Photoelectrochemical Water-Splitting. Energy Chem. 2022, 66, 529–559, doi:10.1016/J.JECHEM.2021.08.038.
  9. Bai, Y.; Li, C.; Liu, L.; Yamaguchi, Y.; Bahri, M.; Yang, H.; Gardner, A.; Zwijnenburg, M.A.; Browning, N.D.; Cowan, A.J.; et al. Photocatalytic Overall Water Splitting Under Visible Light Enabled by a Particulate Conjugated Polymer Loaded with Palladium and Iridium**. Chemie 2022, 134, e202201299, doi:10.1002/ANGE.202201299.
  10. Zhu, H.; Ding, R.; Dou, X.; Zhou, J.; Luo, H.; Duan, L.; Zhang, Y.; Yu, L. Metal Mesh and Narrow Band Gap Mn0.5Cd0.5S Photocatalyst Cooperation for Efficient Hydrogen Production. 2022, Vol. 15, Page 5861 2022, 15, 5861, doi:10.3390/MA15175861.
  11. The findings/novelty of the present work should be clearly highlighted in the revised text.

Reply: We described the findings of the work in a more favorable light in the abstract and conclusion and highlighted them in yellow.

  1. The presented XRD analyses are quite good. The authors can give a more detailed study of the XRD results. They can calculate the lattice parameters of the studied samples, such as crystallite size, lattice strain, and dislocation density of your material, with full discussions. Please also add hkl on the XRD charts. This will give more enhancement efforts to the study.

Reply: Table 1 lists such an important lattice parameter as the volume of a cubic cell. In addition, the influence of the introduction of manganese ions into the crystal lattice of cadmium sulfide is discussed: «As the manganese content increased, the particle size and unit cell volume decreased. Note that the unit cell volume for CdS is equal to 99.79 AÌŠ3 (PDF #41-1049), whereas for MnS this parameter is equal to 88.40 AÌŠ3 (PDF #40-1289).  In addition, there was an insignificant shift of the diffraction peaks toward larger angles 2θ, associated with the smaller radius of Mn2+ compared to Cd2+. Consequently, as x increased in Cd1-xMnxS, the interplanar spacing decreased in the samples». Also, we added into Table 1 the particle size parameters for all phases.

Unfortunately, it is quite difficult to calculate such defectiveness parameters as lattice strain and dislocation density is rather difficult due to the very high defectiveness of the synthesized structures.

Indexes hkl were indicated in XRD patterns of the samples.

  1. The produced materials required some additional investigation characteristics such as PET, FI-SEM, and AFM.

Reply: We added FE-SEM images for the sample 0.1% NiS/Mn0.6 HT120 (Figure S3). Unfortunately, we have been given a very short time for revision and we do not have time to use other methods.

  1. A simple illustration scheme with explanation text for the photocatalytic process (mechanism of action) is highly appreciated to be added to the revised text.

Reply: We added the illustration with a mechanism of heterojunction formation to Figure 7b.

  1. The graphical abstract is also appreciated for the present work, which will give the reader a good feeling about the manuscript content, therefore the authors are highly appreciated for designing a creative interesting graphical abstract for this work. Or by other words, the authors can add it as an illustration scheme inside the discussion part.

Reply: We added the graphical abstract in the manuscript. We have attached it as a separate file, because according to the rules of the journal, you cannot use the figures from the article as a graphic abstract.

  1. Few mistakes are found in the text, the authors are kindly required to recheck the whole revised text for any typos and grammatical errors. 

Reply: We once again checked the text carefully and corrected errors and inaccuracies.

Reviewer 2 Report

The present manuscript Title: “Efficient photocatalytic hydrogen production over NiS-modified cadmium and manganese sulfide solid solutions" deals with the investigation of the Photocatalytic hydogen production based on Cd1-xMnxS sulfide solid solutions were synthesized by varying the fraction of MnS (x = 0.4, 0.6, and 0.8) and the hydrothermal treatment temperature (T = 100, 120, 140, and 160 oC). The introduction and background are given the premise of the manuscript. The results are consistent with the data and figures presented in the manuscript. While I believe this topic is of great interest, I think it needs minor revision before it is ready for publication. So, I recommend this manuscript for publication with minor revisions. Although there are some specific comments listed below.

1) line 93: What do you mean by excess of added Na2S?

2) Images should be provided for readers to read without getting bored.

3) The resolutions of the graphics should be clearer (especially for Fig. 2 and Fig. 3)

4) It should be stated more clearly why a structure with MnS is preferred because of the bad gap energy of MnS in visible light and the decrease in photocatalytic activity due to its formation in the structure.

Author Response

We thank the referee for valuable comments.

1) line 93: What do you mean by excess of added Na2S?

Reply: We mean that twofold excess of Na2S from the stoichiometric amount was added; the explanation was given in the text.

2) Images should be provided for readers to read without getting bored.

Reply: We provided Graphical Abstract and Figure 7b describing the heterojunctions formation.

3) The resolutions of the graphics should be clearer (especially for Fig. 2 and Fig. 3)

Reply: We improved the quality of all figures including XRD patterns. Also, Indexes hkl were indicated in XRD patterns of the samples.

4) It should be stated more clearly why a structure with MnS is preferred because of the bad gap energy of MnS in visible light and the decrease in photocatalytic activity due to its formation in the structure.

Reply: In our opinion, it is the Cd1-xMnxS solid solution structure that is preferable. The solid solution Cd1-xMnxS has a higher activity due to the more negative position of the electrochemical level of the conduction band and therefore higher reduction ability of photogenerated electrons compared to the pristine cadmium sulfide. However, this material still has good sensitivity to visible light. We added the explanation into the text.

Reviewer 3 Report

This work synthesized and modified materials based on Cd1-xMnxS with Pt and NiS cocatalysts. The photocatalytic activity was thoroughly evaluated, and the authors demonstrated in detail the process of hydrogen evolution from Moreover, by evaluating the physical characterization and chemical properties of the materials, especially the rate of hydrogen evolution under visible light (425 nm), the nickel-containing materials can be considered in the future as a replacement for expensive noble metals.

The whole manuscript is very well written and the English grammar is perfect. The basic argumentation process is also adequate, and no errors or corrections can be found. Except for the following three minor flaws, which may need to be improved, I suggest to accept the manuscript directly.

1 Could the clarity of the XRD image be improved, it is blurry compared to the other images.

2 why Na2S/Na2SO3 system and glucose solution were chosen be listed in detail. What are the advantages and disadvantages of these two systems compared to the work of this manuscript in the work of others.

3 Can you add a schematic diagram of the working mechanism.

Author Response

  1. Could the clarity of the XRD image be improved, it is blurry compared to the other images.

Reply: We improved the quality of XRD patterns; Miller indices were also added into the figures.

  1. Why Na2S/Na2SO3 system and glucose solution were chosen be listed in detail. What are the advantages and disadvantages of these two systems compared to the work of this manuscript in the work of others.

Reply: The sulfide system is not only considered as an electron donor, but also affects the catalyst from anodic photocorrosion of the sulfide catalyst. Cadmium sulfide can be easily oxidized by photogenerated holes, and Cd2+ ions from the photocatalyst leach into the aqueous solution. When sodium sulfide is used as a sacrificial agent, dissolved sulfide anions are oxidized instead of lattice sulfide anions in the CdS structure. However, the use of this system of electron donors is not advantageous from a practical point of view.

On the contrary, the use of glucose as a sacrificial agent is of interest from a practical point of view, since glucose can be obtained by hydrolysis of plant biomass components. However, the rate of the hydrogen evolution in the case of the use of this system is lower compared to the Na2S/Na2SO3 electron donor system.

With regard to comparison with other works, we obtained an extraordinary activity in the production of hydrogen from Na2S/Na2SO3 aqueous solutions; the activity of proposed photocatalysts based on Cd1-xMnxS using glucose as a sacrificial agent was investigated for the first time and also has a very high numerical level.

This information was given in the text and highlighted in yellow.

  1. Can you add a schematic diagram of the working mechanism.

Reply: We added the illustration with a mechanism of heterojunction formation to Figure 7.
